# Autoconj: Recognizing and Exploiting Conjugacy Without a Domain-Specific Language

**Matthew D. Hoffman**[*]
Google AI
mhoffman@google.com

**Matthew J Johnson**[*]
Google Brain
mattjj@google.com

**Dustin Tran**
Google Brain
trandustin@google.com

## Abstract

Deriving conditional and marginal distributions using conjugacy relationships can be time consuming and error prone. In this paper, we propose a strategy for automating such derivations. Unlike previous systems which focus on relationships between pairs of random variables, our system (which we call *Autoconj*) operates directly on Python functions that compute log-joint distribution functions. Autoconj provides support for conjugacy-exploiting algorithms in any Python-embedded PPL. This paves the way for accelerating development of novel inference algorithms and structure-exploiting modeling strategies.[1]

## 1 Introduction

Some models enjoy a property called *conjugacy* that makes computation easier. Conjugacy lets us compute *complete conditional* distributions, that is, the distribution of some variable conditioned on all other variables in the model. Complete conditionals are at the heart of many classical statistical inference algorithms such as Gibbs sampling (Geman and Geman, 1984), coordinate-ascent variational inference (Jordan et al., 1999), and even the venerable expectation-maximization (EM) algorithm (Dempster et al., 1977). Conjugacy also makes it possible to marginalize out some variables, which makes many algorithms faster and/or more accurate (e.g.; Griffiths and Steyvers, 2004). Many popular models in the literature enjoy some form of conjugacy, and these models can often be extended in ways that preserve conjugacy.

For experienced researchers, deriving conditional and marginal distributions using conjugacy relationships is straightforward. But it is also time consuming and error prone, and diagnosing bugs in these derivations can require significant effort (Cook et al., 2006).

These considerations motivated specialized systems such as BUGS (Spiegelhalter et al., 1995), VIBES (Winn and Bishop, 2005), and their many successors. In these systems, one specifies a model in a probabilistic programming language (PPL), provides observed values for some of the model's variables, and lets the system automatically translate the model specification into an algorithm (typically Gibbs sampling or variational inference) that approximates the model's posterior conditioned on the observed variables.

These systems are useful, but their monolithic design imposes a major limitation: they are difficult to compose with other systems. For example, a user who wants to interleave Gibbs sampling steps with some customized Markov chain Monte Carlo (MCMC) kernel will find it very difficult to take advantage of BUGS' Gibbs sampler.

In this paper, we propose a different strategy for exploiting conditional conjugacy relationships. Unlike previous approaches (which focus on relationships between pairs of random variables) our

---

[*]equal contribution

[1] Autoconj (including experiments) is available at https://github.com/google-research/autoconj.

system (which we call *Autoconj*) operates directly on Python functions that compute log-joint distribution functions. If asked to compute a marginal distribution, Autoconj returns a Python function that implements that marginal distribution's log-joint. If asked to compute a complete conditional, it returns a Python function that returns distribution objects.

Autoconj is not tied to any particular approximate inference algorithm. But, because Autoconj is a simple Python API, implementing conjugacy-exploiting approximate inference algorithms using Autoconj is easy and fast (as we demonstrate in section 5). In particular, working in the Python/NumPy ecosystem gives Autoconj users access to vectorized kernels, automatic differentiation (via Autograd (Maclaurin et al., 2014)), sophisticated optimization algorithms (via scipy.optimize), and even accelerated hardware (via TensorFlow).

Autoconj provides support for conjugacy-exploiting algorithms in any Python-embedded PPL. More ambitiously, we hope that, just as automatic differentiation has accelerated research in deep learning, Autoconj will accelerate the development of novel inference algorithms and modeling strategies that exploit conjugacy.

## 2   Background: Exponential Families and Conjugacy

To develop a system that can automatically find and exploit conjugacy, we first develop a general perspective on exponential families. Given a probability space $(\mathcal{X}, \mathcal{B}(\mathcal{X}), \nu)$, where $\mathcal{B}(\mathcal{X})$ is the Borel sigma algebra with respect to the standard topology on $\mathcal{X}$, and a statistic function $t : \mathcal{X} \to \mathbb{R}^n$, define the corresponding exponential family of densities (Wainwright and Jordan, 2008), indexed by the natural parameter $\eta \in \mathbb{R}^n$, and log-normalizer function $\mathcal{A}$ as

$$p(x; \eta) = \exp\left\{ \langle \eta,\, t(x) \rangle - \mathcal{A}(\eta) \right\}, \qquad \mathcal{A}(\eta) \triangleq \log \int \exp\left\{ \langle \eta,\, t(x) \rangle \right\} \nu(\mathrm{d}x), \qquad (1)$$

where $\langle \cdot,\, \cdot \rangle$ denotes the standard inner product. The log-normalizer function $\mathcal{A}$ is directly related to the cumulant-generating function, and in particular it satisfies

$$\nabla \mathcal{A}(\eta) = \mathbb{E}\left[ t(x) \right], \qquad \nabla^2 \mathcal{A}(\eta) = \mathbb{E}\left[ t(x) t(x)^{\mathsf{T}} \right] - \mathbb{E}\left[ t(x) \right] \mathbb{E}\left[ t(x) \right]^{\mathsf{T}}, \qquad (2)$$

where the expectation is with respect to $p(x; \eta)$. For a given statistic function $t$, when the corresponding distribution can be sampled efficiently, and when $\mathcal{A}$ and its derivatives can be evaluated efficiently, we say the exponential family (or the statistic function that defines it) is *tractable*.

Consider an exponential-family model where the log density has the form

$$\log p(z, x) = \log p(z_1, z_2, \ldots, z_M, x) = \sum_{\beta \in \boldsymbol{\beta}} \langle \eta_\beta(x),\, t_{z_1}(z_1)^{\beta_1} \otimes \cdots \otimes t_{z_M}(z_M)^{\beta_M} \rangle$$
$$\triangleq g(t_{z_1}(z_1), \ldots, t_{z_M}(z_M)), \qquad (3)$$

where $\beta \subseteq \{0, 1\}^M$ is an index set, we take $t_{z_m}(z_m)^0 \equiv 1$, and where the functions $\{t_{z_m}(z_m)\}_{m=1}^M$ are each the sufficient statistics of a tractable exponential family. In words, the log joint density $\log p(z, x)$ can be written as a multilinear (or multiaffine) polynomial $g$ applied to the statistic functions $\{t_{z_m}(z_m)\}$. These models arise when building complex distributions from simpler, tractable ones, and the algebraic structure in $g$ corresponds to graphical model structure (Wainwright and Jordan, 2008; Koller and Friedman, 2009). In general the posterior $\log p(z \mid x)$ is not tractable, but it admits efficient approximate inference algorithms.

Models of the form (3) are known as *conditionally conjugate* models. Each conditional $p(z_m \mid z_{\neg m})$ (where $z_{\neg m} \triangleq \{z_1, \ldots, z_M\} \setminus \{z_m\}$) is a tractable exponential family. Moreover, the parameters of these conditional densities can be extracted using differentiation. We formalize this below.

**Claim 2.1.** *Given an exponential family with density of the form* (3)*, we have*

$$p(z_m \mid z_{\neg m}) = \exp\left\{ \langle \eta_{z_m}^*,\, t_{z_m}(z_m) \rangle - \mathcal{A}_{z_m}(\eta_{z_m}^*) \right\} \;\; where \;\; \eta_{z_m}^* \triangleq \nabla_{t_{z_m}} g(t_{z_1}(z_1), \ldots, t_{z_M}(z_M)).$$

As a consequence, if we had code for evaluating the functions $g$ and $\{t_{z_m}\}$, along with a table of sampling routines corresponding to each tractable statistic $t_{z_m}$, then we could use automatic differentiation to write a generic Gibbs sampling algorithm. This generic algorithm could be extended to work with any tractable exponential-family distribution simply by populating a table matching

tractable statistics functions to their corresponding samplers. Note this differs from a table of pairs of random variables: conjugacy derives from this lower-level algebraic relationship.

The model structure (3) can be exploited in other approximate inference algorithms, including variational mean field (Wainwright and Jordan, 2008) and stochastic variational inference (Hoffman et al., 2013). Consider the variational distribution

$$q(z) = \prod_m q(z_m; \eta_{z_m}), \qquad q(z_m; \eta_{z_m}) = \exp\left\{\langle \eta_{z_m}, t_{z_m}(z_m)\rangle - \mathcal{A}_{z_m}(\eta_{z_m})\right\}, \qquad (4)$$

where $\eta_{z_m}$ are natural parameters of the variational factors. We write the variational evidence lower bound objective $\mathcal{L} = \mathcal{L}(\eta_{z_1}, \dots, \eta_{z_M})$ for approximating the posterior $p(z \mid x)$ as

$$\log p(x) = \log \int p(z, x)\, \nu_z(\mathrm{d}z) = \log \mathbb{E}_{q(z)}\left[\frac{p(z, x)}{q(z)}\right] \geq \mathbb{E}_{q(z)}\left[\log \frac{p(z, x)}{q(z)}\right] \triangleq \mathcal{L}. \qquad (5)$$

We can write block coordinate ascent updates for this objective using differentiation:

**Claim 2.2.** *Given a model with density of the form* (3) *and variational problem* (4)-(5)*, we have*

$$\arg\max_{\eta_{z_m}} \mathcal{L}(\eta_{z_1}, \dots \eta_{z_M}) = \nabla_{\mu_{z_m}} g(\mu_{z_1}, \dots, \mu_{z_M}) \ \text{where} \ \mu_{z_{m'}} \triangleq \nabla \mathcal{A}_{z_{m'}}(\eta_{z_{m'}}), \ m' = 1, \dots, M.$$

Thus if we had code for evaluating the functions $g$ and $\{t_{z_m}\}$, along with a table of log-normalizer functions $\mathcal{A}_{z_m}$ corresponding to each tractable statistic $t_{z_m}$, then we could use automatic differentiation to write a generic block coordinate-ascent variational inference algorithm. New tractable structures could be added to this algorithm's repertoire simply by populating the table of statistics and their corresponding log-normalizers.

If all this tractable exponential-family structure can be exploited generically, why is writing conjugacy-exploiting inference software still so laborious? The reason is that it is not always easy to get our hands on the representation (3). Even when a model's log joint density *could* be written as in (3), it is often difficult and error-prone to write code to evaluate $g$ directly; it is much more natural to specify model densities without being constrained to this form. The situation is analogous to deep learning research before flexible automatic differentiation: we're stuck writing too much code by hand, and even though in principle this process could be automated, our current software tools aren't up to the task unless we're willing to get locked into a limited mini-language.

Based on this derivation, Autoconj is built to automatically extract these tractable structures (i.e., the functions $g$ and $\{t_{z_m}\}$). It does this given log density functions written in plain Python and NumPy. And it reaps automatic structure-exploiting inference algorithms as a result.

# 3 Analyzing Log-Joint Functions

To extract sufficient statistics and natural parameters from a log-joint function, Autoconj first represent that function in a convenient canonical form. It applies a canonicalization process, which comprises two stages: 1. a tracer maps Python log-joint probability functions to symbolic term graphs; 2. a domain-specific rewrite system puts the log-joint functions in a canonical form and extracts the component functions defined in Section 2.

## 3.1 Tracing Python programs to generate term graphs

The tracer's purpose is to map a Python function denoting a log-joint function to an acyclic term graph data structure. It accomplishes this mapping without having to analyze Python syntax or reason about its semantics directly; instead, the tracer monitors the execution of a Python function in terms of the primitive functions that are applied to its arguments to produce its final output. As a consequence, intermediates like non-primitive function calls and auxiliary data structures, including tuples/lists/dicts as well as custom classes, do not appear in the trace and instead all get traced through. The ultimate output of the tracer is a directed acyclic data flow graph, where nodes represent application of primitive functions (typically NumPy functions) and edges represent data flow. This approach is both simple to implement and able to handle essentially any Python code.

A weakness of this tracing approach is that we only trace one evaluation of the function on example arguments, and we assume that the trace represents the same mathematical function that the original

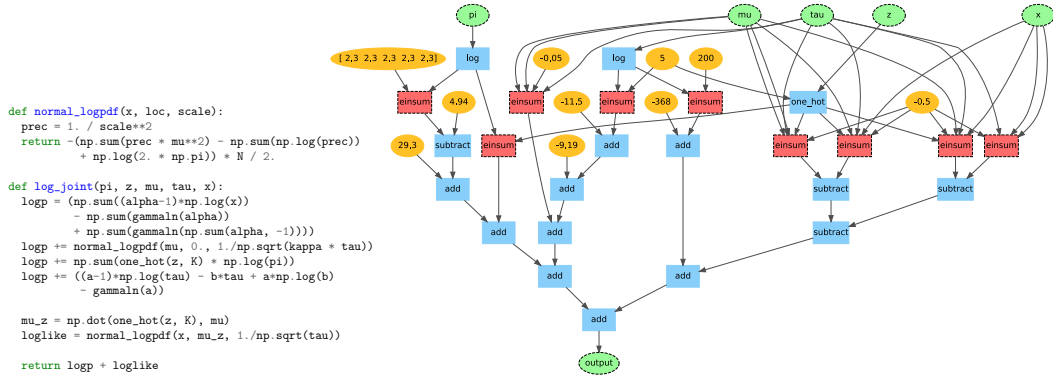

```python
def normal_logpdf(x, loc, scale):
    prec = 1. / scale**2
    return -(np.sum(prec * mu**2) - np.sum(np.log(prec))
            + np.log(2. * np.pi)) * N / 2.

def log_joint(pi, z, mu, tau, x):
    logp = (np.sum((alpha-1)*np.log(x))
            - np.sum(gammaln(alpha))
            + np.sum(gammaln(np.sum(alpha, -1))))
    logp += normal_logpdf(mu, 0., 1./np.sqrt(kappa * tau))
    logp += np.sum(one_hot(z, K) * np.log(pi))
    logp += ((a-1)*np.log(tau) - b*tau + a*np.log(b)
            - gammaln(a))

    mu_z = np.dot(one_hot(z, K), mu)
    loglike = normal_logpdf(x, mu_z, 1./np.sqrt(tau))

    return logp + loglike
```

**Figure 1:** Left: Python code for evaluating the log joint density of a Gaussian mixture model. Right: canonicalized computation graph, representing the same log joint density function but rewritten as a sum of `np.einsum`s of statistic functions.

Python code denotes. This assumption can fail. For example, if a Python function has an if/else that depends on the value of the arguments (and is not expressed in a primitive function), then the tracer could only follow one branch, and so instead raises an error. In the context of tracing log-joint functions, this limitation does not seem to arise too frequently, but it does affect our handling of discrete random variables; for densities of discrete random variables, the tracer can intercept either indexing expressions like `pi[z]` or the use of the primitive function `one_hot`.

Figure 1 summarizes the tracer's use on Python code to generate a term graph. To implement the tracing mechanism, we reuse Autograd's tracer (Maclaurin et al., 2014), which is designed to be general-purpose and extensible with a simple API. Other similar tracing mechanisms are common in probabilistic programming (Goodman and Stuhlmüller, 2014).

## 3.2 Domain-specific term graph rewriting system

The goal of the rewrite system is to take a log-joint term graph and manipulate it into a canonical form. Mathematically, the canonical form described in Section 2 is a multilinear polynomial $g$ on tensor-valued statistic functions $t_1, \ldots, t_M$. For term graphs, we say a term graph is in this canonical form when its output node represents a sum of `np.einsum` nodes, with each `np.einsum` node corresponding to a monomial term in $g$ and each `np.einsum` argument being either a constant, a nonlinear function of an input, or an input itself, with the latter two cases corresponding to statistic functions $t_m$. We rely on `np.einsum` because it is capable of expressing arbitrary tensor contractions, meaning it is a uniform way to express arbitrary monomial terms in $g$.

At its core, the rewrite system is based on pattern-directed invocation of rewrite rules, each of which can match and then modify a small subgraph corresponding to a few primitive function applications. Our pattern language is a new Python-embedded DSL, which is compiled into continuation-passing matcher combinators (Radul, 2013). In addition to basic matchers for data types and each primitive function, the pattern combinators include `Choice`, which produces a match if any of its argument combinators produce a match, and `Segment`, which can match any number of elements in a list, including argument lists. By using continuation passing, backtracking is effectively handled by the Python call stack, and it's straightforward to extract just one match or all possible matches. The pattern language compiler is only ~300 lines and is fully extensible by registering new syntax handlers.

A rewrite rule is then a pattern paired with a rewriter function. A rewriter essentially represents a syntactic macro operating on the term subgraph, using matched sub-terms collected by the pattern to generate a new term subgraph. To specify each rewriter, we again make use of the tracer: we simply write a Python function that, when traced on appropriate arguments, produces the new subgraph, which we then patch into the term graph. This mechanism is analogous to quasiquoting (Radul, 2013), since it specifies a syntactic transformation in terms of native Python expressions. Thus by using pattern matching and tracing-based rewriters, we can define general rewrite rules without writing any code that manually traverses or modifies the term graph data structure. As a result,

it is straightforward to add new rewrite rules to the system. See Listing 1 for an example rewrite rule.

```
pat = (Einsum, Str('formula'), Segment('args1'),
       (Choice(Subtract('op'), Add('op')), Val('x'), Val('y')), Segment('args2'))

def rewriter(formula, op, x, y, args1, args2):
  return op(np.einsum(formula, *(args1 + (x,) + args2)),
            np.einsum(formula, *(args1 + (y,) + args2)))

distribute_einsum = Rule(pat, rewriter)  # Rule is a namedtuple
```

**Listing 1:** A rewrite for distributing `np.einsum` over addition and subtraction.

Rewrite rules are composed into a term rewriting system by an alternating strategy with two steps. In the first step, for each rule we look for a pattern match anywhere in the term graph starting from the output; if no match is found then the process terminates, and if there is a match we apply the corresponding rewriter and move to the second step. In the second step, we traverse the graph from the inputs to the output, performing common subexpression elimination (CSE) and applying local simplifications that only involve one primitive at a time (like replacing a `np.dot` with an equivalent `np.einsum`) and hence don't require pattern matching. By alternating rewrites with CSE, we remove any redundancies introduced by the rewrites. It is straightforward to compose new rewrite systems, involving different sets of rewrite rules or different strategies for applying them.

The process is summarized in Figure 1. The rewriting process aims to transform the term graph of a log joint density into the canonical sum-of-einsums polynomial form corresponding to Eq. (3) (up to commutativity). We do not have a proof that the rewrites are terminating or confluent (Baader and Nipkow, 1999), and the set of possible terms is very complex, though intuitively each rewrite rule applied makes strict progress towards the canonical form (e.g. by distributing multiplication across addition). In practice there have been no problems with termination or normalization.

Once we have processed the log-joint term graph into a canonical form, it is straightforward to extract the objects of interest (namely the statistic functions $t_1, \ldots, t_M$ and the polynomial $g$), match the tractable statistics with corresponding log-normalizer and sampler functions from a table, and perform any further manipulations like automatic differentiation. Moreover, we can map the term graph back into a Python function (via an interpreter), so the rewrite system is hermetic: we can use its output with any other Python tools, like Autograd or SciPy, without those tools needing to know anything about it.

Term rewriting systems have a long history in compilers and symbolic math systems (Sussman et al., 2018; Radul, 2013; Diehl, 2013; Rozenberg, 1997; Baader and Nipkow, 1999). The main novelty here is the application domain and specific concerns and capabilities that arise from it; we're manipulating exponential families of densities for multidimensional random variables, and hence our system is focused on matrix and tensor manipulations, which have limited support in other systems, and a specific canonical form informed by structure-exploiting approximate inference algorithms. Our implementation is closely related to the term rewriting system in `scmutils` (Sussman et al., 2018) and Rules (Radul, 2013), which also use a pattern language (embedded in Scheme) based on continuation-passing matcher combinators and quasiquote-based syntactic macros. Two differences in the implementation are that our system operates on term graphs rather than syntax trees, and that we use tracing to implement a kind of macro system on our term graph data structures (instead of using Scheme's built-in quasiquotes and homoiconicity).

### 3.3 Recognizing Sufficient Statistics and Natural Parameters

Once the log-joint graph has been canonicalized as a sum of `np.einsum`s of functions of the inputs, we can discover and extract exponential-family structure.

Suppose we are interested in the complete conditional of an input $z$. We first need to find all nodes that represent sufficient statistics of $z$. We begin at the output node, and search up through the graph, ignoring any nodes that do not depend on $z$. We walk through any `add` or `subtract` nodes until we reach an `np.einsum` node. If $z$ is a parent of more than one argument of that `np.einsum` node, then the node represents a nonlinear function of $z$ and we label it as a sufficient statistic (if the node

has any inputs that do not depend on $z$ we also need to split those out). Otherwise, we walk through the `np.einsum` node since it is a linear function of $z$. If at any point in the search we reach either $z$ or a node that is not linear in $z$ (i.e., an `add`, `subtract`, or `np.einsum`), we label it as a sufficient statistic.

Once we have found the set of sufficient statistic nodes, we can determine whether they correspond to a known tractable exponential family. For example, in Figure 1, $z$ has integer support and the one-hot statistic, so its complete conditional is a categorical distribution; $\pi$'s support is the simplex and its only sufficient statistic is $\log \pi$, so $\pi$'s complete conditional is a Dirichlet; $\tau$'s support is the non-negative reals, and its sufficient statistics are $\tau$ and $\log \tau$, so its complete conditional is a gamma distribution. If the sufficient-statistic functions do not correspond to a known exponential family, then the system raises an exception.

Finally, to get the natural parameters we can simply take the symbolic gradient of the output node with respect to each sufficient-statistic node using Autograd.

## 4    Related Work

Many probabilistic programming languages (PPLs) exploit conjugacy relationships. PPLs like BUGS (Spiegelhalter et al., 1995), VIBES (Winn and Bishop, 2005), and Augur (Tristan et al., 2014) build an explicit graph of random variables and find conjugate pairs in that graph. This strategy remains widely applicable, but ties the system very strongly to the PPL's model representation. Most recently, Birch (Murray et al., 2018) utilizes a flexible strategy for combining conjugacy and approximate inference in order to enable algorithms such as Sequential Monte Carlo with Rao-Blackwellization. Autoconj could extend their conjugacy component.

PPLs such as Hakaru (Narayanan et al., 2016) have considered treating conditioning and marginalization as program transformations based on computer algebra (Carette and Shan, 2016; Gehr et al., 2016). Unfortunately, most existing computer algebra systems have very limited support for linear algebra and multidimensional array processing, which in turn makes it hard for these systems to either express models using NumPy-style broadcasting or take advantage of vectorized hardware (although Narayanan and Shan (2017) take steps to address this). Exploiting multivariate-Gaussian structure in these languages is particularly cumbersome. Orthogonal to our work, Narayanan and Shan (2017) advances symbolic manipulation for general probability spaces such as mixed discrete-and-continuous events. These ideas could also be used in Autoconj.

## 5    Examples and Experiments

In this section we provide code snippets and empirical results to demonstrate Autoconj's functionality, as well as the benefits of being embedded in Python as opposed to a more narrowly focused domain-specific language. We begin with some examples.

Listing 2 demonstrates doing exact conditioning and marginalization in a trivial Beta-Bernoulli model. The log-joint is implemented using NumPy, and is passed to `complete_conditional()` and `marginalize()`. These functions also take an `argnum` parameter that says which parameter to marginalize out or take the complete conditional of (0 in this example, referring to `counts_prob`) and a `support` parameter. Finally, they take a list of dummy arguments that are used to propagate shapes and types when tracing the log-joint function.

Listing 3 demonstrates how one can handle a more complicated compound prior: the normal-gamma distribution, which is the natural conjugate prior for Bayesian linear regression. Note that we can call `complete_conditional()` on the function produced by `marginalize()`.

We can extend the marginalize-and-condition strategy above to more complicated models. In the supplement, we demonstrate how one can implement the Kalman-filter recursion with Autoconj. The generative process is

$$x_1 \sim \text{Normal}(0, s_0); \quad x_{t>1} \sim \text{Normal}(x_{t-1}, s_x); \quad y_t \sim \text{Normal}(x_t, s_y). \tag{6}$$

The core recursion consists of using `marginalize()` to compute $p(x_{t+1}, y_{t+1} \mid y_{1:t})$ from the functions $p(x_t \mid y_{1:t})$ and $p(x_{t+1}, y_{t+1} \mid x_t)$, then using `marginalize()` again to compute

```python
def log_joint(counts_prob, n_heads, n_draws, prior_a, prior_b):
    log_prob = (prior_a-1)*np.log(counts_prob) + (prior_b-1)*np.log1p(-counts_prob)
    log_prob += n_heads*np.log(counts_prob) + (n_draws-n_heads)*np.log1p(-counts_prob)
    log_prob += -gammaln(prior_a) - gammaln(prior_b) + gammaln(prior_a + prior_b)
    return log_prob

n_heads, n_draws = 60, 100
prior_a, prior_b = 0.5, 0.5
all_args = [0.5, n_heads, n_draws, prior_a, prior_b]
make_complete_conditional = autoconj.complete_conditional(
    log_joint, 0, SupportTypes.UNIT_INTERVAL, *all_args)
# A Beta(60.5, 40.5) distribution object.
complete_conditional = make_complete_conditional(n_heads, n_draws, prior_a, prior_b)
# Computes the marginal log-probability of n_heads, n_draws given prior_a, prior_b
marginal = autoconj.marginalize(log_joint, 0, SupportTypes.UNIT_INTERVAL, *all_args)
print('log p(n_heads=60 | a, b) =', marginal(n_heads, n_draws, prior_a, prior_b))
```

**Listing 2:** Exact inference in a simple Beta-Bernoulli model.

```python
def log_joint(tau, beta, x, y, a, b, kappa, mu0):
    log_p_tau = log_probs.gamma_gen_log_prob(tau, a, b)
    log_p_beta = log_probs.norm_gen_log_prob(beta, mu0, 1. / np.sqrt(kappa * tau))
    log_p_y = log_probs.norm_gen_log_prob(y, np.dot(x, beta), 1. / np.sqrt(tau))
    return log_p_tau + log_p_beta + log_p_y

# log p(tau, x, y), marginalizing out beta
tau_x_y_log_prob = autoconj.marginalize(log_joint, 1, SupportTypes.REAL, *all_args)
# compute and sample from p(tau | x, y)
make_tau_posterior = autoconj.complete_conditional(
    tau_x_y_log_prob, 0, SupportTypes.NONNEGATIVE, *all_args_ex_beta)
tau_sample = make_tau_posterior(x, y, a, b, kappa, mu0).rvs()
# compute and sample from p(beta | tau, x, y)
make_beta_conditional = autoconj.complete_conditional(
    log_joint, 1, SupportTypes.REAL, *all_args)
beta_sample = make_beta_conditional(tau, x, y, a, b, kappa, mu0)
```

**Listing 3:** Exact inference in a Bayesian linear regression with normal-gamma compound prior. We factorize the joint posterior on the mean and precision as $p(\mu, \tau \mid x, y) = p(\tau \mid x, y)p(\mu \mid x, y, \tau)$. We first compute the *marginal* joint distribution $p(x, y, \tau)$ by calling `marginalize()` on the full log-joint. We then compute the marginal posterior $p(\tau \mid x, y)$ by calling `complete_conditional()` on the marginal $p(x, y, \tau)$, and finally we compute $p(\mu \mid x, y, \tau)$ by calling `complete_conditional()` on the full log-joint.

$p(y_{t+1} \mid y_{1:t})$ and `complete_conditional()` to compute $p(x_{t+1} \mid y_{1:t+1})$. As in the normal-gamma example, it is up to the user to reason about the graphical model structure, but Autoconj handles all of the conditioning and marginalization automatically. The same code could be applied to a hidden Markov model (which has the same graphical model structure) by simply changing the distributions in the log-joint and the support from real to integer.

When not all complete conditionals are tractable, the variational evidence lower bound (ELBO) is not tractable to compute exactly. Several strategies exist for dealing with this problem. One approach is to find a lower bound on the log-joint that is only a function of expected sufficient statistics of some exponential family (Jaakkola and Jordan, 1996; Blei and Lafferty, 2005). Another is to linearize problematic terms in the log-joint (Khan et al., 2015).

Knowledge of conjugate pairs is not sufficient to implement either of these strategies, which rely on direct manipulation of the log-joint to achieve a kind of quasi-conjugacy. But Autoconj naturally facilitates these strategies, since it does not require that the log-joint functions it is given exactly correspond to any true generative process.

Listing 4 demonstrates variational inference for Bayesian logistic regression (which has a non-conjugate likelihood) using Autoconj to optimize the bound of Jaakkola and Jordan (1996). One

```python
def log_joint_bound(beta, xi, x, y):
  log_prior = np.sum(-0.5 * beta**2 - 0.5 * np.log(2*np.pi))
  y_logits = (2 * y - 1) * np.dot(x, beta)
  # Lower bound on -log(1 + exp(-y_logits)).
  lamda = (0.5 - expit(xi)) / (2. * xi)
  log_likelihood_bound = np.sum(-np.log(1 + np.exp(-xi)) + 0.5 * (y_logits - xi)
                                + lamda * (y_logits ** 2 - xi ** 2))
  return log_prior + log_likelihood_bound

def xi_update(beta_mean, beta_secondmoment, x):
  """Sets the bound parameters xi to their optimal value."""
  beta_cov = beta_secondmoment - np.outer(beta_mean, beta_mean)
  return np.sqrt(np.einsum('ij,ni,nj->n', beta_cov, x, x) +
                 x.dot(beta_mean)**2)

neg_energy, (t_beta,), (lognorm_beta,), = meanfield.multilin_repr(
    log_joint_bound, argnums=(0,), supports=(SupportTypes.REAL,),
    example_args=(beta, xi, x, y))
elbo = partial(meanfield.elbo, neg_energy, (lognorm_beta,))
mu_beta = grad(lognorm_beta)(grad(neg_energy)(t_beta(beta), xi, x, y))   # initialize

for iteration in range(100):
  xi = xi_update(mu_beta[0], mu_beta[1], x)
  mu_beta = grad(lognorm_beta)(grad(neg_energy)(mu_beta, xi, x, y))
  print('{}\t{}'.format(iteration, elbo(mu_beta, xi, x, y)))
```

**Listing 4:** Variational Bayesian logistic regression using the lower bound of Jaakkola and Jordan (1996). Autoconj can work with `log_joint_bound()` even though it is not a true log-joint density.

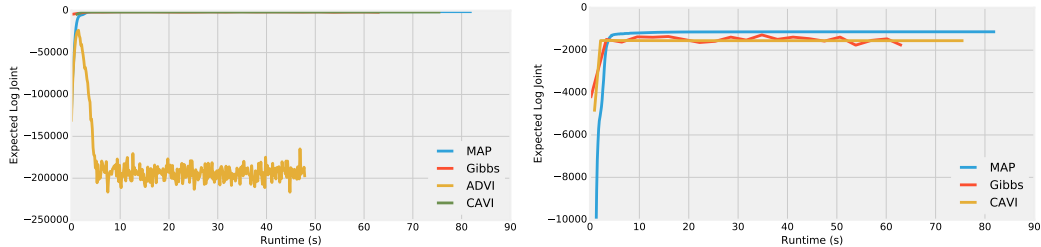

**Figure 2:** Comparison of algorithms for Bayesian factor analysis according to their estimate of the expected log-joint as a function of runtime. **(left)** Relative to other algorithms, mean-field ADVI grossly underfits. **(right)** Zoom-in on other algorithms. Block coordinate-ascent variational inference (CAVI) converges faster than Gibbs.

could also use Autoconj to implement other methods such as proximal variational inference (Khan and Wu, 2017; Khan et al., 2016, 2015).

**Factor Analysis**   Autoconj facilitates many structure-exploiting inference algorithms. Here, we demonstrate why such algorithms are important for efficient inference, and that Autoconj supports their diverse collection. We generate data from a linear factor model,

$$w_{mk} \sim \text{Normal}(0,1); \; z_{nk} \sim \text{Normal}(0,1); \; \tau \sim \text{Gamma}(\alpha,\beta); \; x_{mn} \sim \text{Normal}(w_m^\top z_n, \tau^{-1/2}).$$

There are $N$ examples of $D$-dimensional vectors $x \in \mathbb{R}^{N \times D}$, and the data assumes a latent factorization according to all examples' feature representations $z \in \mathbb{R}^{N \times K}$ and the principal components $w \in \mathbb{R}^{D \times K}$. As a toy demonstration, we use relatively small $N$, $D$, and $K$.

Autoconj naturally produces a structured mean-field approximation, since conditioned on $w$ and $x$ the rows of $z$ each have multivariate-Gaussian complete conditionals (and vice versa for $z$ and $w$). We compared Autoconj structured block coordinate-ascent variational inference (CAVI) with Autoconj block Gibbs, mean-field ADVI (Kucukelbir et al., 2016), and MAP implemented using scipy.optimize. All algorithms besides ADVI yield reasonable results, demonstrating the value of exploiting conjugacy when it is available.

| Implementation | Runtime (s) |
|---|---|
| **Autoconj (NumPy; 1 CPU)** | **62.9** |
| **Autoconj (TensorFlow; 1 CPU)** | **75.9** |
| **Autoconj (TensorFlow; 6 CPU)** | **19.7** |
| **Autoconj (TensorFlow; 1 GPU)** | **4.3** |

**Table 1:** Time to run 500 iterations of variational inference on a mixture of Gaussians. TensorFlow offers little advantage on one CPU core, but an order-of-magnitude speedup on GPU.

**Benchmarking Autoconj**   While we used NumPy as a numerical backend for Autoconj, other Python-based backends are possible. We wrote a simple translator that replaces NumPy ops in our computation graph to TensorFlow ops (Abadi et al., 2016). We can therefore take a log-joint written in NumPy, extract complete conditionals or marginals from that model, and then run the conditional or marginal computations in a TensorFlow graph (possibly on a GPU or TPU).

We ran Autoconj's CAVI in NumPy and TensorFlow for a mixture-of-Gaussians model:

$$\pi \sim \text{Dirichlet}(\alpha); \quad z_n \sim \text{Categorical}(\pi); \quad \mu_{kd} \sim \text{Normal}(0, \sigma); \quad \tau_{kd} \sim \text{Gamma}(a, b);$$

$$x_{nd} \sim \text{Normal}(\mu_{z_n d}, \tau_{z_n d}^{-1/2}).$$

See Listing 5. We automatically translated the NumPy CAVI ops to TensorFlow ops, and benchmarked 500 iterations of CAVI in NumPy and TensorFlow on CPU and GPU. Table 1 shows the results, which clearly demonstrate the value of running on GPUs.

```python
import autoconj.pplham as ph   # a simple "probabilistic programming language"

def make_model(alpha, beta):
  def sample_model():
    """Generates matrix of shape [num_examples, num_features]."""
    epsilon = ph.norm.rvs(0, 1, size=[num_examples, num_latents])
    w = ph.norm.rvs(0, 1, size=[num_features, num_latents])
    tau = ph.gamma.rvs(alpha, beta)
    x = ph.norm.rvs(np.dot(epsilon, w.T), 1. / np.sqrt(tau))
    return [epsilon, w, tau, x]
  return sample_model

num_examples = 50
num_features = 10
num_latents = 5
alpha = 2.
beta = 8.
sampler = make_model(alpha, beta)

log_joint_fn = ph.make_log_joint_fn(sampler)
```

**Listing 5:** Implementing the log joint function for Table 1. This example also illustrates how Autoconj could be embedded in a probabilistic programming language where models are sampling functions and utilities exist for tracing their execution (e.g., Tran et al. (2018)).

## 6   Discussion

In this paper, we proposed a strategy for automatically deriving conjugacy relationships. Unlike previous systems which focus on relationships between pairs of random variables, Autoconj operates directly on Python functions that compute log-joint distribution functions. This provides support for conjugacy-exploiting algorithms in any Python-embedded PPL. This paves the way for accelerating development of novel inference algorithms and structure-exploiting modeling strategies.

**Acknowledgements.** We thank the anonymous reviewers for their suggestions and Hung Bui for helpful discussions.

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
