[Supplementary Material]

# Autoconj: Recognizing and Exploiting Conjugacy Without a Domain-Specific Language Supplemental Material

**Matthew D. Hoffman**[*]
Google AI
mhoffman@google.com

**Matthew J Johnson**[*]
Google Brain
mattjj@google.com

**Dustin Tran**
Google Brain
trandustin@google.com

## 1 Code Examples

### 1.1 Kalman Filter

Listing 1 demonstrates computing the marginal likelihood of a time-series $y_{1:T}$ under the linear-Gaussian model

$$x_1 \sim \text{Normal}(0, s_0); \quad x_{t>1} \sim \text{Normal}(x_{t-1}, s_x); \quad y_t \sim \text{Normal}(x_t, s_y). \tag{1}$$

```
def log_p_x1_y1(x1, y1, x1_scale, y1_scale):
  """Computes log p(x_1, y_1)."""
  log_p_x1 = log_probs.norm_gen_log_prob(x1, 0, x1_scale)
  log_p_y1_given_x1 = log_probs.norm_gen_log_prob(y1, x1, y1_scale)
  return log_p_x1 + log_p_y1_given_x1

def log_p_xt_xtt_ytt(xt, xtt, ytt, xt_prior_mean, xt_prior_scale, x_scale,
                     y_scale):
  """Given log p(x_t | y_{1:t}), computes log p(x_t, x_{t+1}, y_{t+1})."""
  log_p_xt = log_probs.norm_gen_log_prob(xt, xt_prior_mean, xt_prior_scale)
  log_p_xtt = log_probs.norm_gen_log_prob(xtt, xt, x_scale)
  log_p_ytt = log_probs.norm_gen_log_prob(ytt, xtt, y_scale)
  return log_p_xt + log_p_xtt + log_p_ytt

def make_marginal_fn():
  # p(x_1 | y_1)
  x1_given_y1_factory = complete_conditional(
      log_p_x1_y1, 0, SupportTypes.REAL, *([1.] * 4))
  # log p(y_1)
  log_p_y1 = marginalize(log_p_x1_y1, 0, SupportTypes.REAL, *([1.] * 4))

  # Given p(x_t | y_{1:t}), compute log p(x_{t+1}, y_{t+1} | y_{1:t}).
  log_p_xtt_ytt = marginalize(
      log_p_xt_xtt_ytt, 0, SupportTypes.REAL, *([1.] * 7))
  # Given p(x_{t+1}, y_{t+1} | y_{1:t}), compute log p(y_{t+1} | y_{1:t}).
  log_p_ytt = marginalize(
      log_p_xtt_ytt, 0, SupportTypes.REAL, *([1.] * 6))
  # Given p(x_{t+1}, y_{t+1} | y_{1:t}), compute p(x_{t+1} | y_{1:t+1}).
  xt_conditional_factory = complete_conditional(
      log_p_xtt_ytt, 0, SupportTypes.REAL, *([1.] * 6))

  def marginal(y_list, x_scale, y_scale):
    # Initialization: compute log p(y_1), p(x_1 | y_1).
    log_p_y = log_p_y1(y_list[0], x_scale, y_scale)
    xt_conditional = x1_given_y1_factory(y_list[0], x_scale, y_scale)

    for t in range(1, len(y_list)):
      # Compute log p(y_t | y_{1:t-1}).
      log_p_y += log_p_ytt(y_list[t], xt_conditional.args[0],
                           xt_conditional.args[1], x_scale, y_scale)
```

---

[*]equal contribution

```
    # Compute p(x_t | y_{1:t}).
    xt_conditional = xt_conditional_factory(
        y_list[t], xt_conditional.args[0], xt_conditional.args[1], x_scale,
        y_scale)
  return log_p_y
return marginal
```

**Listing 1:** Exact marginalization in a Kalman filter.