[Reviews · NeurIPS 2018]

Reviewer 1



The paper proposes a tracing and rewriting system called Autoconj that automatically exploits conjugacy of exponential distributions in probabilistic programs. The tool operates on ordinary Python programs without using a domain-specific language. The authors provide a brief description of how the rewriting systems works and demonstrate how to use it on several examples. The paper is well written and the contribution is potentially significant. As noted by the authors, conjugate relationships were previously exploited within probabilistic programming, but only within a context of domain-specific languages. This made it very difficult to incorporate such tools within more generic PPLs, since that would require compiling the probabilistic program into the required DSL, applying the simplification, and compiling back. Since Autoconj operates directly on Python programs, it should be possible to easily deploy it within another Python-based PPL. I should point out, however, that this claim was not demonstrated in the paper. The submission would be very strong if it included an example of using Autoconj within an existing PPL that doesn't currently exploit conjugacy. I am not an expert on exponential families so perhaps I'm missing something obvious but I found section 2 a little confusing. Specifically, are claims 2.1 and 2.2 original contributions or are they known results? If the former I would expect them to be proven, at least in the appendix, even if such proofs are considered straightforward. If the latter I would expect a specific pointer to existing literature. My main complaint about the paper is that the rewriter is only described at a high level in section 3. The description gives me some idea how it works but nowhere near enough for me to implement such a tool. A table of rewrite rules or even a small calculus on which the rewrites can be formally defined would go a long way here. The authors claim their rewriting algorithms always terminates producing a result in a canonical form but I have no way of verifying that statement if the rules aren't actually presented. The examples are well chosen and useful for understanding how Autoconj can be applied in practice, in particular showing that it can indeed be applied to normal Python programs. The experiment with factor analysis demonstrates that exploiting conjugacy can be very useful for approximate inference. The second experiment shows that running code on GPU and be much faster than on CPU and the authors point out that the machinery needed to implement Autoconj can be very easily extended to convert NumPy operations to Tensorflow getting extra performance for free. Overall I believe the Autoconj tool can be very useful but I'm not sure if the paper itself provides that much value. It feels a bit like reading the extended abstract, where the underlying methods are mentioned but not discussed in depth. I don't think there is enough detail in there for others to implement a similar system. I still recommend acceptance though, since for better or worse this level of description of software systems seems to be more or less standard in the community.

Reviewer 2



This paper shows that prior-likelihood conjugacy relationships can be recognized and exploited from a Python program that computes a log-probability in a form such as the sum of a log-prior and a log-likelihood. The core of the approach is read out the log-joint expression (by operator overloading) then rewrite it into a canonical form (which can then be recognized). This work makes substantial progress on an important problem. The crucial and novel part of this work is the system of term-writing rules and the canonical form that emerges, not how term read-out and rewriting are implemented in Python. Unfortunately, the paper does not describe the rewriting rules (in the form of a table, say) or the canonical form (in the form of a grammar, say). The paper should give enough information to justify the central claim (lines 174-176) that "the rewriting process is guaranteed to terminate". As it stands, this claim is not justified: why is it that "our rewrite rules do not generate cycles" and what is the mathematical definition for a rewrite to "make progress toward a canonical form"? An unorganized pile of rewriting rules is not a robust or scalable way to perform computer algebra, so it's essential that the paper explain how this pile of rewriting rules is in fact organized. It seems that Autoconj can produce both distribution objects (which can draw samples) as well as density functions. Because the prior and the posterior related by conjugacy are usually represented in the same way, I expected Autoconj to be able to take distribution objects as input also, so I was disoriented to find that the input to Autoconj is a density in all examples. Please clarify. The examples in Section 5 and the discussion of related work in Section 4 should be better related. In particular, the paper should clarify whether Autoconj can "exploit multivariate-Gaussian structure" -- how is the model on lines 319-320 programmed, and is the Kalman filter in the supplemental material multivariate-Gaussian? I read the author response. I encourage the author(s) to revise the paper (not just the appendix) to explain (not just specify) the rewriting rules and their termination. The high-level description given in the author response makes it seem as if the rewriting rules merely expand a polynomial into a sum of monomials. So in particular, the sequence of metric-decreasing rewrites that handle the Kalman filter in the supplemental material should be made completely explicit. I also encourage the author(s) to clarify the setting of the paper by addressing Reviewer #3's questions in the paper.

Reviewer 3



In this paper, the authors propose a system, called Autoconj, which is able to recognize and exploit conjugacy by analyzing Python functions that compute log-joint distribution functions. Autoconj is expected to facilitate the inference of probabilistic models. It looks like Autoconj has potential to facilitate inference of the probabilistic model, which might be very useful. The writing of the paper looks structured, but I have difficulty in understanding some details in the paper. I do not understand the contribution of Autoconj. It looks like Autoconj takes a python function, which represents the log-joint distribution function as the input. Does this log-joint distribution function need to be properly normalized? What is the output of Autoconj? What does it mean by “recognizing and exploiting conjugacy”? Does it mean that Autoconj is able to automatically generate the marginal and conditional distribution according to the query? If so, how are these distributions represented? Are these distributions represented in terms of a black-box pdf (i.e., given a query for a certain latent variable $z_1 = c$, Autoconj returns the approximated posterior pdf value $q(c; \eta_{z_1})$), or it returns the form and the parameters of the posterior distribution (i.e., Autoconj returns that the posterior distribution of $z_1$ is a Gaussian distribution parameterized by a specific mean and variance, for example)? In the title, it is mentioned that Autoconj is “without a Domain-Specific Language”. What does “domain-specific” mean? Why is it important to a system like Autoconj? I also have difficulty in understanding some details about Autoconj. In Equation (3), the variable $z$ are involved. Are these variables represent the latent variables in the probabilistic model? In Equation (4) and (5), it looks like Autoconj is based on variational inference. But in line 38, it says “Autoconj is not tied to any particular approximate inference algorithms”. So is Autoconj based on variational inference approximation? Does the “canonical form” mentioned in line 130 mean Equation (3)? In Section 3.3, Autoconj analyzes the sufficient statistics for the posterior distribution. Since the joint function is not provided in a symbolic form, how does Autoconj know which statistic is relevant? How the process described in Section 3 related to the equations described in Section 2? In addition, I typically expect that a NIPS paper involves development of new models and theorems. It looks like Autoconj is an implementation of existing models, as described in Section 2. I am also not sure whether this paper is interesting to the NIPS community. In summary, Autoconj might be potentially useful because it facilitates inference of probabilistic models. I cannot fully understand the paper, and am not sure whether this paper would interest NIPS community. Based on the information that I can understand about this paper, I do not suggest acceptance of this paper. I have read the author feedback. I am still not very certain how Autoconj is implemented, especially for the rewriting system. Probably more details should be included. In addition, the contribution of the paper is still not clear to me. I still do not know how this method is different from existing systems. I do not think implementing the method in python itself is a very important contribution.